# Comparisons of Clinical Outcomes in Women with Advanced Ovarian Cancer Treated with Frontline Intraperitoneal versus Dose-Dense Platinum/Paclitaxel Chemotherapy without Bevacizumab

**DOI:** 10.3390/ijerph17103603

**Published:** 2020-05-20

**Authors:** Wan-Hua Ting, Chi-Huang Hsiao, Hui-Hua Chen, Ming-Chow Wei, Ho-Hsiung Lin, Sheng-Mou Hsiao

**Affiliations:** 1Department of Obstetrics and Gynecology, Far Eastern Memorial Hospital, New Taipei 220409, Taiwan; stellatingwh@gmail.com (W.-H.T.); thandaaye24@gmail.com (H.-H.C.); wei@mail.femh.org.tw (M.-C.W.); hhlin@ntuh.gov.tw (H.-H.L.); 2Division of Medical Oncology and Hematology, Department of Internal Medicine, Far Eastern Memorial Hospital, New Taipei 220409, Taiwan; hsiao.frank.ch@gmail.com; 3Department of Obstetrics and Gynecology, National Taiwan University College of Medicine and National Taiwan University Hospital, Taipei 100225, Taiwan; 4Graduate School of Biotechnology and Bioengineering, Yuan Ze University, Taoyuan 320315, Taiwan

**Keywords:** ovarian neoplasms, chemotherapy, adjuvant, cytoreduction, peritoneum

## Abstract

*Background:* We aimed to compare the clinical outcomes between intraperitoneal chemotherapy and dose-dense chemotherapy for the frontline treatment of advanced ovarian, fallopian tube and primary peritoneal cancer in women not receiving bevacizumab. *Methods:* All consecutive women with stage II~IV cancer treated with either frontline intraperitoneal or dose-dense platinum/paclitaxel chemotherapy and not receiving bevacizumab between March 2006 and June 2019 were reviewed. *Results:* A total of 50 women (intraperitoneal group, *n* = 22; dose-dense group, *n* = 28) were reviewed. Median progression-free survival (32.6 months versus 14.2 months; adjusted hazard ratio = 0.38; 95% CI = 0.16 to 0.90, *p* = 0.03) and overall survival (not reached versus 30.7 months; adjusted hazard ratio = 0.23, 95% CI = 0.07 to 0.79, *p* = 0.02) were significantly higher in the intraperitoneal group than in the dose-dense group. A multivariable Cox proportional-hazards model also indicated that the number of frontline chemotherapy cycles (adjusted hazard ratio = 0.66, 95% CI 0.47 to 0.94, *p* = 0.02) was a predictor of better overall survival. Nausea/vomiting and nephrotoxicity occurred more frequently in the intraperitoneal group (*p* = 0.02 and <0.0001, respectively). *Conclusions:* Intraperitoneal chemotherapy seems to be superior in progression free survival and overall survival to dose-dense chemotherapy in the frontline treatment of women with optimally resected advanced ovarian, fallopian tube or primary peritoneal cancer and not receiving bevacizumab.

## 1. Introduction

The combination of paclitaxel and carboplatin is the standard first-line chemotherapy for ovarian, fallopian tube and primary peritoneal cancer and is conventionally given via the intravenous route every 3 weeks. Several trials have demonstrated a clinically significant survival advantage associated with intraperitoneal (IP) chemotherapy compared to intravenous chemotherapy, and the best outcomes have consistently been seen in women who have no residual disease [1,2,3]. Landrum et al. reported a median overall survival (OS) of 110 months in stage III women with no residual disease who received IP chemotherapy [4]. Later, in year 2009, the Japanese Gynecologic Oncology Group (JGOG) demonstrated the superiority of triweekly carboplatin and dose-dense weekly paclitaxel in improving the progression-free survival (PFS, median: 28.2 vs.17.5 months; hazard ratio = 0.76, *p* = 0.0037) and OS of women with stage II–IV ovarian cancer (median: 100.5 vs. 62.2 months; hazard ratio = 0.79, *p* = 0.039) compared with the conventional 21-day regimen [5,6].

No direct prospective comparisons of clinical outcomes were made between IP and dose-dense chemotherapy until the most recent report released by the Gynecologic Oncology Group (GOG) protocol 252, which showed no significant difference in the survival rates between the groups in women with stage II–IV optimally resected disease; nonetheless, it is worthwhile to mention that all women in the study received bevacizumab [7]. Bevacizumab is expensive and its cost might not be covered by the national health insurance in some countries, thus the issue of optimal postoperative adjuvant chemotherapy (IP vs. dose-dense) without the addition of bevacizumab for women with stage II–IV ovarian cancer remains important. To the best of our knowledge, there is only one literature comparing the therapeutic effect of IP vs. dose-dense chemotherapy in the frontline treatment of women with advanced ovarian cancer and not receiving bevacizumab, and there was no between-group difference in OS between the IP and dose-dense chemotherapy groups [8]. Thus, we were interested and aimed to elucidate whether in the frontline treatment of advanced ovarian, fallopian tube and primary peritoneal cancer, IP chemotherapy would confer an improved survival benefit compared with dose-dense chemotherapy among women not receiving bevacizumab.

## 2. Materials and Methods

From March 2006 to May 2019, the medical records of all consecutive women aged 20 and above with International Federation of Gynecology and Obstetrics (FIGO) stage II–IV advanced ovarian, fallopian tube or primary peritoneal cancer who received postoperative or neoadjuvant IP or dose-dense platinum/paclitaxel chemotherapy in a tertiary referral center were reviewed. IP chemotherapy was defined as having one or more cycles of an IP regimen administered. The decision for route of chemotherapy administration was based on physician’s discretion. Since its first introduction to our institution on April 2014, IP chemotherapy has been widely used in clinical setting. This study was approved by the Research Ethics Review Committee of the hospital (NO. 108137-E).

The IP regimen was given as follows: 135 mg/m^2^ intravenous paclitaxel over a 3 or 24 h period on day 1, followed by 75–100 mg/m^2^ IP cisplatin on day 2 and 60 mg/m^2^ IP paclitaxel on day 8 [3]. For women with significantly impaired renal function (i.e., estimated glomerular filtration rate < 50 mL/min/1.73 m^2^), carboplatin (area under the curve [AUC] = 6) was used instead of cisplatin. The dose-dense regimen was given as intravenous carboplatin at a dose calculated to produce an AUC of 6 mg/mL per min on day 1, followed by 80 mg/m^2^ intravenous paclitaxel on days 1, 8, and 15 [5]. The dose of carboplatin was calculated with the formula of Calvert and colleagues [9]. The treatments were repeated every 3 weeks for six cycles. Those women without achievement of complete response after 6 cycles of chemotherapy might be treated with an additional 1–2 cycles of chemotherapy.

Optimal debulking surgery was defined as having residual tumor with a maximal diameter less than 1 cm after cytoreductive surgery; the others were defined as suboptimal debulking surgery.

Toxicity was graded according to the National Cancer Institute Common Terminology Criteria for Adverse Events. Radiologic assessment included chest-X ray and abdominopelvic computed tomography (CT) scan. Treatment response was evaluated according to the World Health Organization criteria [10]. In patients with a cancer antigen-125 (CA-125) ≥ 40 IU/mL and without measurable tumors, response and progression were evaluated according to Rustin’s criteria [11,12]. A complete response (CR) was defined as the disappearance of all evidence (including CA-125 and image) of tumor for at least 4 weeks. A CR could not be determined by CA-125 alone. A partial response (PR) was defined as a ≥50% reduction in the products of each measurable lesion or a ≥50% reduction of CA-125 for at least 4 weeks. Progressive disease was defined as a ≥25% increase in the size of one or more measurable lesions, the appearance of new lesions, or a ≥25% increase of CA-125. Stable disease was defined as any condition not meeting any of the above criteria [11,12,13].

Disease recurrence was assessed according to the CA-125 criteria of disease progression [11,12], the appearance of abnormal radiological findings, or histological proof from biopsy analyses, whichever occurred first.

OS was calculated as the time interval from the date of surgery to the date of death from any cause or the last follow-up. PFS was defined as the time interval from the date of surgery to clinically defined recurrence, disease progression, or the last follow-up. Stata version 11.0 (Stata Corp, College Station, TX, USA) was used for statistical analyses. Survival curves were generated using the Kaplan–Meier method, and differences in the survival curves were calculated with the log-rank test. A *p*-value less than 0.05 was considered statistically significant. Multivariable Cox proportional hazards model was used to identify independent predictors of PFS and OS. Receiver operating characteristic (ROC) curve analysis was performed to identify optimal cutoff values. The optimal cutoff value was determined by the point on the ROC curve that was closest to the upper left-hand corner.

## 3. Results

A total of 50 consecutive women who underwent debulking surgery and adjuvant IP (*n* = 22) or dose-dense (*n* = 28) chemotherapy with platinum/paclitaxel were reviewed (Table 1). None of them received bevacizumab throughout the course. There were no significant differences in the baseline characteristics between the two groups of women (Table 1).

Median PFS (32.6 months versus 14.2 months, *p* = 0.03, Figure 1a, Table 1) and OS (not reached versus 30.7 months, *p* = 0.03, Figure 1b, Table 1) were significantly higher in the IP group than in the dose-dense group.

Multivariable Cox proportional-hazards model revealed that IP chemotherapy was the only independent predictor of PFS (hazard ratio = 0.38, 95% confidence interval (CI) = 0.16 to 0.90, *p* = 0.03, Table 2).

Multivariable Cox proportional-hazards model also revealed that IP chemotherapy was an independent predictor of OS (hazard ratio = 0.23, 95% CI = 0.07 to 0.79, *p* = 0.02, Table 3). In addition, the number of IP or dose-dense chemotherapy cycles (hazard ratio = 0.66, 95% CI = 0.47 to 0.94, *p* = 0.02) was another independent predictor of OS (Table 3).

For women who underwent optimal debulking surgery (*n* = 25), PFS (*p* = 0.01, Figure 1c) was significantly different between the IP and dose-dense groups. Nonetheless, there was a trend to be statistically significant in OS between the IP and dose-dense groups (*p* = 0.052, Figure 1d).

For women who underwent suboptimal debulking surgery (*n* = 20), there were no between-group differences in PFS (*p* = 0.61, Figure 1e) and OS (*p* = 0.21, Figure 1f).

The number of IP or dose-dense chemotherapy cycles ≥ 6 was determined to be the optimum cut-off value to predict death using ROC analysis, which provided an area under the ROC curve of 0.52 (95% CI = 0.40 to 0.63; sensitivity = 91.3%, specificity = 22.2%, Figure 2).

The toxicity profile is summarized in Table 4. A significantly higher proportion of women in the IP group suffered from nausea/vomiting and nephrotoxicity compared with those in the dose-dense chemotherapy group (Table 4). Further analysis did not reveal any between-group differences in grade 3 and 4 adverse events between the IP and dose-dense treatment groups (*p* = 0.12).

## 4. Discussion

Our study demonstrated both PFS and OS advantages of IP chemotherapy in women with advanced ovarian, fallopian tube or primary peritoneal cancer compared with dose-dense chemotherapy. Similarly, Rettenmaier et al. observed prolonged PFS in women treated with IP chemotherapy compared to those treated with dose-dense chemotherapy (34.0 vs. 27.6 months). Despite the lack of between-group OS benefit as described in the text of their results (i.e., 42.2 months in the IP group vs. 41.6 months in the dose-dense group) [8], if we extrapolated OS from their Figure 2 [8], it seems that there was a significant between-group difference in their OS (median: approximately 63 months in the IP group vs. 38 months in the dose-dense group) [14], which is similar to our data in Figure 1b and Table 1.

In our study, none of the women received bevacizumab, and we found a survival benefit in both PFS and OS in the IP group, compared with the dose-dense group. Similarly, Bixel et al. excluded women who received bevacizumab with primary therapy and found that IP chemotherapy improved PFS, compared with intravenous chemotherapy [15]. Nonetheless, in women receiving bevacizumab, the GOG-252 study found no survival benefit in the IP group, compared with the dose-dense group [7]. Thus, the above discrepancy in survival benefit between our and the GOG-252 studies might be related to the addition of bevacizumab.

In the JGOG-3016 study, none of the women received bevacizumab; a survival benefit was found in the dose-dense group, compared with the 21-day group [6]. Similarly, in the GOG-262 study, there was a significant difference in PFS between the dose-dense and 21-day groups in women not receiving bevacizumab (14.2 months vs. 10.3 months, *p* = 0.03); nonetheless, in women receiving bevacizumab, there was no between-group difference in PFS (14.9 months vs. 14.7 months, *p* = 0.60) [16]. From these observations, it is reasonable to speculate that the addition of bevacizumab may diminish the superiority of IP chemotherapy, and this might partly explain the discrepancy in survival between our and the GOG-252 studies [7].

The potential clinical benefit of dose-dense chemotherapy might be attributable to the sustained exposure of tumor cells to the cytotoxic effect of chemotherapeutic drugs, and thus reduces the non-exposure interval during which re-growth and neo-angiogenesis occur [17,18]. On the other hand, intraperitoneal chemotherapy provides pharmacologic advantage by directly exposing tumor to a greater concentration of chemotherapeutic drugs [19]. It is unclear which method of delivery is more superior. The result from GOG-252 is debatable due to several confounding factors, including the addition of bevacizumab and modified IP cisplatin regimen [7]. An ongoing clinical trial by the JGOG is comparing 21-day IP vs. intravenous carboplatin in combination with intravenous dose-dense paclitaxel without bevacizumab [20]. We are looking forward to clarifying the confounding role of bevacizumab in interpreting the data from GOG-252.

Emerging evidence suggests the role of antitumor immune response induced by chemotherapy. A study by Chang et al. suggested that dose-dense administration of low dose platinum agent and paclitaxel results in the generation of antitumor immunity by sparing the immune system from major toxicity and modifying the tumor microenvironment in favor of immunogenic tumor cell death [21]. Zunino et al. observed that hyperthermic intraperitoneal chemotherapy not only kills cancer cells but also induces an efficient anticancer immune response that leads to immunogenic death [22]. Whether the route of administration confers any immunological difference remains unknown; a larger clinical study is warranted in future research.

The number of chemotherapy cycles has a pronounced effect on the OS of women with advanced disease who received either IP or dose-dense chemotherapy. Yen et al. reported that at least five cycles are needed to effectively prolong the survival rate in women treated with IP chemotherapy [23], which is similar to our result which showed the number of chemotherapy cycles as an independent predictor of OS (Table 3). From our ROC analysis, the number of IP or dose-dense chemotherapy cycles ≥ 6 was the optimum cutoff value to predict death (Figure 2).

In our study, 18 of 22 (81.8%) women completed six or more cycles of IP chemotherapy. Two women switched early on to dose-dense chemotherapy at the second cycle due to intolerable abdominal pain (*n* = 1) and vaginal leakage of peritoneal fluid (*n* = 1). However, the completion rates were 42%, 60% and 65.1% in the GOG-172 study, Landrum et al.’s study and Yen et al.’s study, respectively [3,4,24]. The main reason for the discontinuation of therapy might be catheter-related complications [25]. In our institution, although we did experience a few IP catheter-related complications, such as port-site infection, malfunction, leakage or dislodgement, as well as intolerable chemotherapy-related side effects such as severe abdominal pain or nausea/vomiting, our insurance system allowed inpatient supportive medical care. These might explain why a high percentage of women underwent ≥ 6 cycles of IP chemotherapy. We believe that good supportive care is the key element to improve the success of IP therapy.

Nausea/vomiting, nephrotoxicity and neurotoxicity were significantly more frequent in the IP group than in the dose-dense group (Table 4). However, there were no between-group differences in grade 3 and 4 adverse events in our study. Thus, IP chemotherapy should remain a good alternative for women with advanced ovarian cancer compared with dose-dense chemotherapy.

The role of IP chemotherapy in women with sub-optimally debulked (≥1 cm residual) disease remains questionable. IP chemotherapy was recommended for women with optimally debulked (<1 cm residual) disease [3]. Nonetheless, Nagao et al. reported that IP carboplatin and intravenous paclitaxel were effective and safe in sub-optimally debulked patients, with a median PFS and OS of 24 and 31 months, respectively [26]. In our study, IP chemotherapy was associated with a better PFS in women with optimal cytoreduction (Figure 1c). In addition, IP chemotherapy seems to have a better survival probability in OS in suboptimal debulking cases (Figure 1f) compared with dose-dense chemotherapy despite a lack of statistical significance (*p* = 0.21).

We acknowledge that the clinical evidence of this study is limited due to its retrospective, limited sample size and nonrandomized nature. The retrospective nature of this study has made it impossible in our multivariate analysis to exclude the confounding effect of some important clinicopathological predictors such as grade, R0 resection and comorbidities. Nevertheless, our data indicate that IP chemotherapy potentially confers a greater survival benefit compared to dose-dense chemotherapy, with tolerable side effects, especially women not receiving bevacizumab.

## 5. Conclusions

IP chemotherapy seems be superior in survival to dose-dense chemotherapy as the frontline treatment for women with optimally resected advanced ovarian, fallopian tube or primary peritoneal cancer and not receiving bevacizumab. The number of frontline chemotherapy cycles was a predictor of better overall survival.

## Figures and Tables

**Figure 1 ijerph-17-03603-f001:**
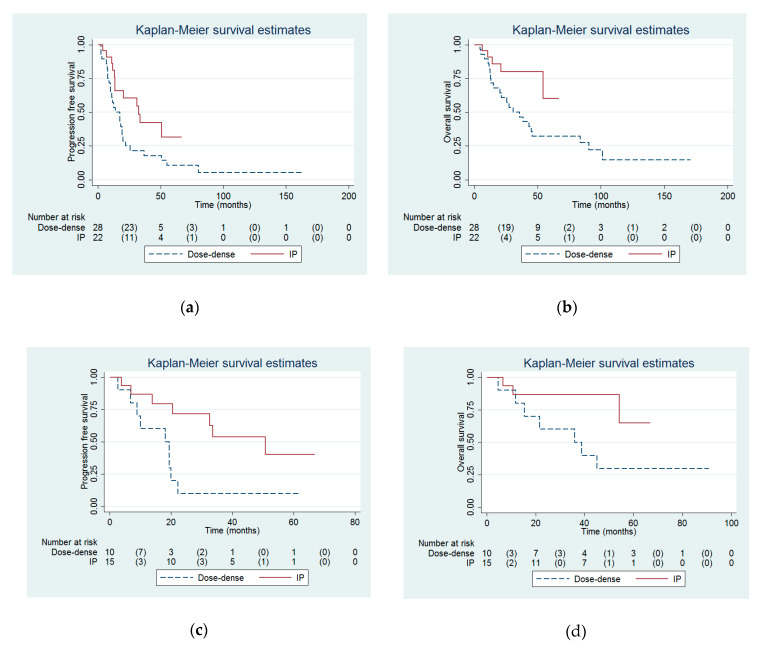
Probabilities of (**a**) progression free survival and (**b**) overall survival between the intraperitoneal (IP) and dose-dense groups in all enrolled women (*n* = 50). Probabilities of (**c**) progression free survival and (**d**) overall survival between the IP and dose-dense groups in optimally debulked women (*n* = 25). Probabilities of (**e**) progression free survival and (**f**) overall survival between the IP and dose-dense groups in sub-optimally debulked women (*n* = 25).

**Figure 2 ijerph-17-03603-f002:**
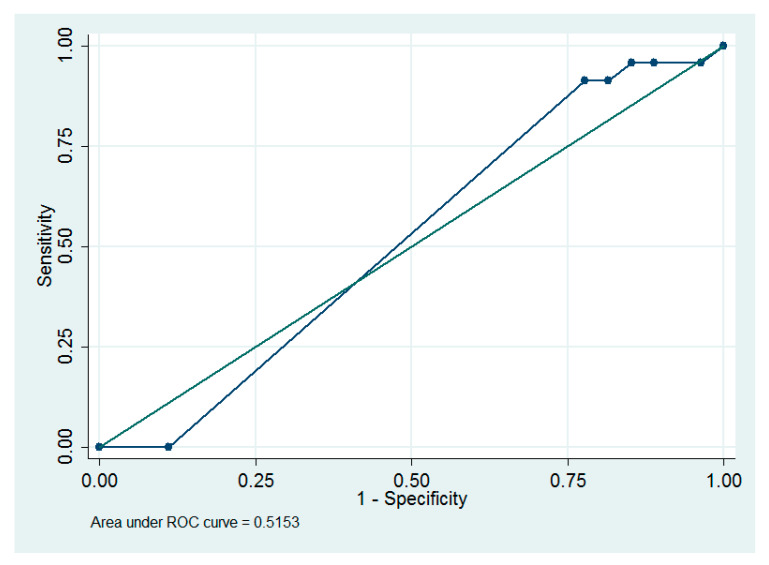
The receiver operating characteristic curve for the number of IP or dose-dense chemotherapy cycles as a predictor of death.

**Table 1 ijerph-17-03603-t001:** Baseline characteristics of women with advanced ovarian, fallopian tube or primary peritoneal cancer (*n* = 50).

Variable	IP (*n* = 22)	Dose-Dense (*n* = 28)	^†^ *p*
Age (years)	52.6 ± 7.4	54.2 ± 8.7	0.59
Body mass index (kg/m^2^)	25.4 ± 3.5	24.2 ± 4.4	0.20
Baseline CA-125 (U/mL)	2120 ± 2665	1180 ± 3330	0.53
Site			
Ovary	19 (86)	28 (100)	0.08
Fallopian tube	1 (5)	0 (0)	
Peritoneum	2 (9)	0 (0)	
FIGO stage			
2	2 (9)	2 (7)	1.00
3	14 (64)	17 (61)	
4	6 (27)	9 (32)	
Histologic subtype			
Serous	14 (61)	18 (66)	1.00
Endometrioid	2 (13)	3 (10)	
Clear cell	3 (13)	3 (10)	
Mucinous	0 (0)	1 (3)	
Others	3 (13)	3 (7)	
Debulking surgery			
^‡^ Optimal	15 (68)	10 (36)	0.10
^‡^ Suboptimal	7 (32)	13 (46)	
ECOG Score			
0	2 (9)	0 (0)	0.09
1	17 (77)	18 (64)	
2	3 (14)	10 (36)	
Lymph node metastasis	13 (59)	19 (68)	0.52
Neoadjuvant chemotherapy	1 (4)	1 (3)	1.00
Number of chemotherapy cycles	5.5 ± 1.6	5.7 ± 1.3	0.59
Progression free interval (months)	32.6 (13.8 to infinity)	14.2 (10 to 19.4)	0.03
Overall survival (months)	Not reached (54.3 to infinity)	30.7 (15.4 to 46)	0.03
Clinical response			
CR	19 (86)	18 (64)	0.31
PR	1 (5)	5 (18)	
SD	0 (0)	1 (4)	
PD	2 (9)	4 (14)	
Follow-up interval (months)	33.4 ± 18.3	48.3 ± 30.9	0.60
Progression or recurrence	12 (55)	26 (93)	0.002
Subsequent therapy	8 (36)	21 (75)	0.006
Subsequent therapy lines	1.4 ± 0.9	2.4 ± 1.4	0.04

Values are expressed as mean ± standard deviation or number (percentage). BMI = body mass index, CA-125 = cancer antigen 125, CI = confidence interval, CR = complete response, ECOG = Eastern Cooperative Oncology Group, FIGO = The International Federation of Gynecology and Obstetrics, IP = intraperitoneal, PD = progressive disease, PR = partial response, SD = stable disease. ^†^ By Wilcoxon rank-sum test, chi-square test or Fisher’s exact test. ^‡^ Data about residual tumor after primary surgery was missing in 5 cases of the dose-dense group.

**Table 2 ijerph-17-03603-t002:** Cox proportional-hazards model to predict progression free survival (*n* = 50).

Variable	Univariate			Multivariable		
	Hazard Ratio	95% CI	^†^ *p*	Hazard Ratio	95% CI	^‡^ *p*
Regimen (IP = 1 vs. dose-dense = 0)	0.47	0.24 to 0.94	0.03	0.38	0.16 to 0.90	0.03
Age (years)	1.02	0.98 to 1.07	0.27	1.01	0.97 to 1.06	0.60
FIGO stage	1.92	1.08 to 3.39	0.03	0.97	0.41 to 2.27	0.95
ECOG score	1.76	0.96 to 3.23	0.07	1.19	0.49 to 2.93	0.70
Suboptimal debulking	2.18	1.09 to 4.32	0.03	1.53	0.65 to 3.67	0.34
Number of IP or dose-dense chemotherapy cycles	0.91	0.73 to 1.14	0.41	0.78	0.59 to 1.04	0.09

CA-125 = cancer antigen 125, CI = confidence interval, ECOG = Eastern Cooperative Oncology Group, FIGO = The International Federation of Gynecology and Obstetrics, IP = intraperitoneal. ^†^ Univariate Cox proportional-hazards model. ^‡^ Multivariable Cox proportional-hazards model.

**Table 3 ijerph-17-03603-t003:** Cox proportional-hazards model to predict overall survival (*n* = 50).

Variable	Univariate			Multivariable		
	Hazard Ratio	95% CI	^†^ *p*	Hazard Ratio	95% CI	^‡^ *p*
Regimen (IP = 1 vs. dose-dense = 0)	0.35	0.13 to 0.93	0.04	0.23	0.07 to 0.79	0.02
Age (years)	1.04	0.99 to 1.10	0.10	1.04	0.99 to 1.10	0.13
FIGO stage	1.34	0.70 to 2.55	0.37	0.57	0.22 to 1.49	0.26
ECOG score	1.55	0.76 to 3.14	0.23	1.33	0.49 to 3.57	0.57
Suboptimal debulking	1.96	0.85 to 4.53	0.12	1.53	0.57 to 4.14	0.40
Number of IP or dose-dense chemotherapy cycles	0.84	0.66 to 1.09	0.19	0.66	0.47 to 0.94	0.02

Abbreviations are the same as Table 2. ^†^ Univariate Cox proportional-hazards model. ^‡^ Multivariable Cox proportional-hazards model.

**Table 4 ijerph-17-03603-t004:** Toxicity profile of women receiving intraperitoneal or dose-dense chemotherapy (*n* = 50).

Grade	Dose-Dense (*n* = 28)				IP (*n* = 22)				^†^ *p*
	1	2	3	4	1	2	3	4	
Hematological									
Leukopenia	3 (11)	12 (43)	12 (43)	1 (4)	1 (5)	8 (36)	8 (36)	2 (9)	0.31
Thrombocytopenia	5 (18)	6 (21)	2 (7)	1 (4)	6 (27)	7 (32)	1 (5)	0 (0)	0.73
Anemia	2 (7)	17 (61)	9 (32)	0 (0)	2 (9)	15 (68)	4 (18)	0 (0)	0.56
Non-hematological									
Nausea/vomiting	17 (61)	4 (14)	1 (4)	0 (0)	8 (36)	10 (45)	4 (18)	0 (0)	0.02
Diarrhea	6 (21)	0 (0)	0 (0)	0 (0)	3 (14)	2 (9)	0 (0)	0 (0)	0.29
Nephrotoxicity	8 (29)	0 (0)	0 (0)	0 (0)	3 (14)	12 (55)	0 (0)	0 (0)	<0.0001
Hepatotoxicity	10 (36)	2 (7)	0 (0)	1 (4)	8 (36)	0 (0)	2 (9)	0 (0)	0.43
Neurotoxicity	12 (43)	4 (14)	0 (0)	0 (0)	6 (27)	10 (45)	0 (0)	0 (0)	0.07

Values are expressed as number (percentage). IP = intraperitoneal. ^†^ By Chi-square test or Fisher’s exact test.

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
