# Peer review of "Comparisons of Clinical Outcomes in Women with Advanced Ovarian Cancer Treated with Frontline Intraperitoneal versus Dose-Dense Platinum/Paclitaxel Chemotherapy without Bevacizumab"

_ijerph, 2020, doi:10.3390/ijerph17103603_

Round 1
Reviewer 1 Report
Comments,
The manuscript by Hsiao co-workers concerns comparisons of clinical outcomes in women with advanced ovarian cancer treated with frontline intraperitoneal versus dose-dense platinum/paclitaxel chemotherapy without Bevacizumab. The authors carefully studied the comparison of the clinical outcomes between intraperitoneal chemotherapy and dose-dense chemotherapy for the frontline treatment of advanced ovarian, fallopian tube, and primary peritoneal cancer in women. Overall the manuscript is written well, and scientifically strong. Therefore, the present manuscript is suitable for IJERPH
Thank
Author Response
Ms. Ref. No.: ijerph-792560
Title: Comparisons of Clinical Outcomes in Women with Advanced Ovarian Cancer Treated with Frontline Intraperitoneal versus Dose-dense Platinum/Paclitaxel Chemotherapy without Bevacizumab
Response to the reviewers’ comments:
Reviewer #1:
- The manuscript by Hsiao co-workers concerns comparisons of clinical outcomes in women with advanced ovarian cancer treated with frontline intraperitoneal versus dose-dense platinum/paclitaxel chemotherapy without Bevacizumab. The authors carefully studied the comparison of the clinical outcomes between intraperitoneal chemotherapy and dose-dense chemotherapy for the frontline treatment of advanced ovarian, fallopian tube, and primary peritoneal cancer in women. Overall the manuscript is written well, and scientifically strong. Therefore, the present manuscript is suitable for IJERPH
Answer: Thank you. Your comment is very much appreciated.

Reviewer 2 Report
Well written case series.
It is not specified on what basis were patients assigned for IP or IV groups.
Apart from notions of survival curves, no exact data are given about follow-up times, recurrence-rates, frequency and types of subsequent treatment lines of the two groups.
The promising survival benefit in the IP group over that in the IV group is clearly related to the better surgical results. Unfortunately, the small case numbers and the retrospective nature of the study has made it impossible in the multivariate analysis to exclude the confounding effect of important clinicopathological predictors such as grade, histological type, lymph node status, R0 resection and comorbidities. The conclusions could only be supported if additional proof was given regarding the similarity of the extent and result (i.e. the size and location of residual disease) of debulking in the compared groups.
The presented data proved the survival advantage for the IP vs IV group only in optimally debulked cases (Figure 1) but not in suboptimally resected cases (Figure 2). Therefore, the conclusion should at least be changed as "Intraperitoneal chemotherapy seems to be superior in progression free survival and overall survival to dose-dense chemotherapy in the frontline treatment of women with optimally resected advanced ovarian, fallopian tube or primary peritoneal cancer and not receiving bevacizumab".
Author Response
Ms. Ref. No.: ijerph-792560
Title: Comparisons of Clinical Outcomes in Women with Advanced Ovarian Cancer Treated with Frontline Intraperitoneal versus Dose-dense Platinum/Paclitaxel Chemotherapy without Bevacizumab
Response to Reviewer 2 comments:
- It is not specified on what basis were patients assigned for IP or IV groups.
Answer: Thank you very much for your comment. Dose-dense chemotherapy has been practiced in our institution as early as in year 2009, intraperitoneal chemotherapy was later introduced in year 2014. The type of adjuvant therapy assigned was at physician’s discretion. We have added a few sentences in the Material and Methods section (please refer to page 2, line 68-71 in the revised manuscript).
- Apart from notions of survival curves, no exact data are given about follow-up times, recurrence-rates, frequency and types of subsequent treatment lines of the two groups.
Answer: Thank you very much for your comment. We have added relevant data in the bottom of Table 1 for your reference (please refer to page 4, Table 1 in the revised manuscript).
- The promising survival benefit in the IP group over that in the IV group is clearly related to the better surgical results. Unfortunately, the small case numbers and the retrospective nature of the study has made it impossible in the multivariate analysis to exclude the confounding effect of important clinicopathological predictors such as grade, histological type, lymph node status, R0 resection and comorbidities. The conclusions could only be supported if additional proof was given regarding the similarity of the extent and result (i.e. the size and location of residual disease) of debulking in the compared groups.
Answer: Thank you very much for your comment. We have added a statement of limitation in the Discussion section (please refer to page 8, line 236-238).
- The presented data proved the survival advantage for the IP vs IV group only in optimally debulked cases (Figure 1) but not in suboptimally resected cases (Figure 2). Therefore, the conclusion should at least be changed as "Intraperitoneal chemotherapy seems to be superior in progression free survival and overall survival to dose-dense chemotherapy in the frontline treatment of women with optimally resected advanced ovarian, fallopian tube or primary peritoneal cancer and not receiving bevacizumab".
Answer: Thank you very much for your comment. We have revised the sentences in the abstract and the main text as suggested (please refer to page 1, line 32 and page 8, line 243 in the revised manuscript).

Reviewer 3 Report
Ting WH et al reported a retrospective study in which compared the clinical outcomes between dose-dense chemotherapy and intraperitoneal chemotherapy in advance ovarian cancer patients not treated with bevacizumab. Results showed a better PFS and OS in those patients treated with IP chemotherapy and underwent to optimal debulking surgery.
The analyses reported by Ting et al are particularly relevant for those patients who cannot be treated with bevacizumab, although the administration of bevacizumab remains the standard of care for advance ovarian cancer due to its relevant impact on the immunological system.
Although the analysis was carried out in a small number of patients, it was methodically well conducted
Do the authors have an explanation that could justify the different outcomes of the two therapeutic approaches? Which is impact on the immunological status of these two methods?The Discussion could be improved by highlighting this aspect.
Author Response
Ms. Ref. No.: ijerph-792560
Title: Comparisons of Clinical Outcomes in Women with Advanced Ovarian Cancer Treated with Frontline Intraperitoneal versus Dose-dense Platinum/Paclitaxel Chemotherapy without Bevacizumab
Response to Reviewer 3 comments:
- Ting WH et al reported a retrospective study in which compared the clinical outcomes between dose-dense chemotherapy and intraperitoneal chemotherapy in advance ovarian cancer patients not treated with bevacizumab. Results showed a better PFS and OS in those patients treated with IP chemotherapy and underwent to optimal debulking surgery. The analyses reported by Ting et al are particularly relevant for those patients who cannot be treated with bevacizumab, although the administration of bevacizumab remains the standard of care for advance ovarian cancer due to its relevant impact on the immunological system. Although the analysis was carried out in a small number of patients, it was methodically well conducted
Do the authors have an explanation that could justify the different outcomes of the two therapeutic approaches? Which is impact on the immunological status of these two methods? The Discussion could be improved by highlighting this aspect.
Answer: Thank you very much for your comment. We have done some research on onco-immunology and added two paragraphs in the Discussion (please refer to page 8, line 188-205).

Round 2
Reviewer 2 Report
The criticisms have been addressed adequately and appropriate corrections have been made to the original version.